# Integrating In Vitro Biopharmaceutics into Physiologically Based Biopharmaceutic Model (PBBM) to Predict Food Effect of BCS IV Zwitterionic Drug (GSK3640254)

**DOI:** 10.3390/pharmaceutics15020521

**Published:** 2023-02-03

**Authors:** Konstantinos Stamatopoulos, Paola Ferrini, Dung Nguyen, Ying Zhang, James M. Butler, Jon Hall, Nena Mistry

**Affiliations:** 1Biopharmaceutics, DPD, MDS, GlaxoSmithKline, David Jack Centre, Park Road, Ware SG12 0DP, UK; 2Analytical Platform and Platform Modernisation, Analytical Development, DPD, MDS, GlaxoSmithKline, Gunnels Wood Road, Stevenage SG1 2NY, UK; 3IVIVT DMPK Research, GlaxoSmithKline, 1250 S. Collegeville Road, Collegeville, PA 19426, USA; 4Clinical Pharmacology Modeling and Simulation, GSK, Collegeville, PA 19426, USA; 5Analytical Development, MDS, GlaxoSmithKline, David Jack Centre, Park Road, Ware SG12 0DP, UK

**Keywords:** PBPK modelling, PBBM modelling, biopharmaceutics, food effect, BCS IV, biorelevant media, in vitro, dissolution, permeability, MDCK cell line, FaSSIF, FeSSIF, Simcyp^®^

## Abstract

A strategy followed to integrate in vitro solubility and permeability data into a PBBM model to predict the food effect of a BCS IV zwitterionic drug (GSK3640254) observed in clinical studies is described. The PBBM model was developed, qualified and verified using clinical data of an immediate release (IR)-tablet (10–320 mg) obtained in healthy volunteers under fasted and fed conditions. The solubility of GSK3640254 was a function of its ionization state, the media composition and pH, whereas its permeability determined using MDCK cell lines was enhanced by the presence of mixed micelles. In vitro data alongside PBBM modelling suggested that the positive food effect observed in the clinical studies was attributed to micelle-mediated enhanced solubility and permeability. The biorelevant media containing oleic acid and cholesterol in fasted and fed levels enabled the model to appropriately capture the magnitude of the food effect. Thus, by using Simcyp^®^ v20 software, the PBBM model accurately predicted the results of the food effect and predicted data were within a two-fold error with 70% being within 1.25-fold. The developed model strategy can be effectively adopted to increase the confidence of using PBBM models to predict the food effect of BCS class IV drugs.

## 1. Introduction

GSK3640254 (GSK254) is a maturation inhibitor demonstrated to inhibit all HIV-1 subtypes with efficacy against a broad range of polymorphisms [1]. In previous clinical studies, GSK254 was orally administrated after a moderate-fat meal (approximately 600 calories with approximately 30% of calories from fat) [2]. The reason for administration with food was based on previous experience with low bioavailability and dose-proportionality concerns observed with GSK3532795, a structurally similar compound [2]. GSK254 is a BCS IV zwitterionic, high lipophilic (LogP 5.49, in-house unpublished data) drug and practically insoluble in water; thus, administration with food to increase its bioavailability was necessary.

Unlike BCS I-III drugs, poorly soluble and low permeable (BCS class IV) drugs pose significant challenges for an effective oral administration, as their absorption depends on whether the rate-limiting step is dissolution or permeability [3].

In the case of zwitterionic drugs, their solubility in biorelevant media is an interplay between the ionization of the drug within the physiological pH range on the gastrointestinal tract (GI tract), and the charge (zeta potential) of the micelles [4]. Takács-Novák, K., et al. [4] showed that for niflumic acid, a BCS III class ampholyte, the predicted food effect (FeSSIF-to-FaSSIF solubility ratio, SFeSSIF/SFaSSIF) was minimal (0.47), although this drug had the highest LogP value (4.81) among the other drugs examined in their work. These findings contradict the clinical data where the exposure of niflumic acid was increased four- and fivefold after administration of its prodrug with low- and high-fat meals, respectively [5]. Thus, the food effect of niflumic acid cannot be explained simply with the biorelevant media used. The typical FaSSIF and FeSSIF media used in their study oversimplify the luminal composition of the GI tract, e.g., use of single bile salt (mainly taurocholate) vs. a mixture of bile salts present in vivo, absence of digestion products (e.g., fatty acids and mono-glycerides) as well as the solubility measured at a single pH value. Knowing that the pH and composition of the media will affect the partitioning of an ampholyte drug to micelles and hence its solubility, biorelevant media reflecting the human intestinal aspirates should be developed/used and the solubility should be measured across a range of pH values. Furthermore, the presence of digestion products, e.g., oleic acid, will change the zeta potential of the micelles to lower negative values, further altering drug–micelles electrostatic interactions. Thus, in this work, the solubility of GSK254 was measured in biorelevant media containing fatty acids and a different mixture of bile salts to reflect population differences in the composition of human duodenal aspirates [6,7] at different pH values.

Unlike BCS class III ampholyte drugs, BCS class IV ampholyte drugs, such as GSK254 with low permeability, will be more susceptible to drug–micelles interaction deriving from changes in media composition and pH. This is of particular importance when the drug is moving from the bulk luminal environment and passing to the mucus, the first barrier for permeation, which is a well-buffered microenvironment with pH ranging from 6.5 to 7.4 [8]. Thus, the interactions of an ampholyte BCS class IV with the micelles might be different between the bulk luminal environment and the mucus.

Previous studies on the use of biorelevant media for transport experiments in the Caco-2 model showed that the permeability, especially of lipophilic drugs, was significantly decreased in the presence of micelles [9]. This was attributed to the reduced free fraction of the drug with increasing the lipid content of the media, i.e., FaSSIF vs. FeSSIF. Thus, the permeability of GSK254 was measured in MDCK cell lines from the presence of biorelevant media to assess how its permeability will be affected. The permeability of GSK254 was also measured at different pH values to determine the impact of ionization.

Although these in vitro exercises might help to understand some critical aspects for the in vivo oral dissolution and absorption of GSK254, this still is a simple approach relative to the complex environment of the human GI tract. Therefore, what would be the clinical relevance of the in vitro results?

In a recently published paper by the IQ consortium, comprising industrial experts in biopharmaceutics and PBPK/PBBM modelling, an industry perspective on the use of PBPK modelling to predict drug–food interactions was discussed [10]. In this paper, the performance of PBPK/PBBM models to predict food effects was assessed across different BCS class drugs with known food effects from the collected clinical studies. The authors did not find a clear trend in prediction success for negative or positive food effects as well as no clear relationship to the BCS class of tested drug substances. However, the authors pointed out that the confidence in the PBPK modelling to predict the food effect could be associated with changes in the GI luminal fluids or physiology, including micellar entrapment, bile salts, fluid volume, motility and pH. For those BCS class IV drugs where the food effect was associated with bile salts and phospholipids solubilization, there was a high confidence in using PBPK modelling to predict the food effect. However, the BCS class (II)/IV drugs selected were either non-ionized in intestinal pH (e.g., aprepitant) or neutral across the physiological pH in the GI tract (e.g., ivacaftor) and possessed moderate to high permeability. Thus, the bile salt mediated solubility enhancement was the main mechanism behind the food effect, which is an “easy” task for the PBBM models to capture the food effect. 

However, zwitterionic drugs are a more complex case for the reasons described above. Considering some limitations of the current PBBM models, a strategy of how to integrate in vitro data to overcome some of these limitations are presented in this work.

There were some key questions to ask to enable data-driven decisions within the model:(1)Should the free or the total luminal concentration be used as reference for permeation?The current assumption is that only the free monomers will permeate the cell membrane. However, what evidence/criteria should enable this decision?(2)Which biorelevant media should be used, and which better reflect the in vivo solubility of GSK254?(3)PBPK platforms, such as Simcyp^®^, use the water-to-micelle partition coefficient (LogK_M:W_) to estimate the total solubility of the drug, as well as to calculate the free fraction available for absorption in the mechanistic permeability model (MechPeff). However, the LogK_M:W_ is a single value applied on two different environments, the bulk luminal environment and the mucus microenvironment. Considering the challenges with different ionization of an ampholyte between the two environments, can a global/single parameter be used and describe both, luminal solubility, and permeability? The reason is that the LogK_M:W_ for the ionized species is a single value and cannot describe the electrostatic repulsion and attraction interactions, occurring interchangeably, between the drug and the micelles. Thus, the luminal pH may favor electrostatic attraction forces, whereas the mucus pH favors repulsion forces and vice versa.

The strategy presented here was followed to integrate the in vitro data into the PBBM model to predict the food effect of GSK254 observed in the clinical studies.

## 2. Materials and Methods

### 2.1. List of Chemicals

Ethanol (Merck (Darmstadt, Germany), absolute ≥99.8% GC), sodium taurochenodeoxycholate (TCDC; Sigma), sodium taurocholate hydrate (TC; Sigma, ≥97.0% TLC), sodium glycodeoxy-cholate (GDC; Sigma, BioXtra, ≥97% HPLC), Trifluoroacetic acid (TFA) 100% (for HPLC, HiPerSolv CHROMANORM, VWR Chemicals, Radnor, PA, USA), sodium glycocholate hydrate (GC; Sigma, ≥95% TLC), cholesterol (CH; Sigma, Sigma grade ≥ 99%), oleic acid (OA; ≥99% GC), L-alpha- phosphatidylcholine from egg yolk (PC; Sigma, type XVI-E ≥ 99% TLC, lyophilized powder), potassium chloride (Alfa Aesar, ACS, 99.0–100.5%), sodium dihydrogenphosphate (Sigma Aldrich, Darmstadt, Germany), sodium acetate trihydrate (Sigma Aldrich, ACS reagent ≥ 99%) and acetic acid (glacial; Alfa Aesar (Ward Hill, MA, USA), 99+%) were used as received. A total of 1 M NaOH solution and 0.1 M HCl solution were prepared to adjust the pH of the buffers.

### 2.2. Solubility Measurements

Instrumentation: UnchainedLab (Freeslate) CM3 multifunctional platform; Mettler Toledo Quantos QB5 automated solid dispensing platform; Thermo Scientific (London, UK) Lynx 4000 centrifuge; Agilent Infinity (San Diego, CA, USA) II 1260 HPLC.

Blank buffer preparation: blank FeSSIF buffers at pH 6.5, 7.4 and 8 were prepared with sodium dihydrogen phosphate and potassium chloride to obtain 0.1 M phosphate concentration and 0.2 M ionic strength. The pH was adjusted with 1.0 M NaOH and 0.1 M HCl. Blank FeSSIF buffer at pH 5 was prepared using sodium acetate trihydrate, KCl and acetic acid to obtain 0.1 M acetate concentration and 0.2 M ionic strength. The pH was measured at 5.01 without further adjustments. Blank FaSSIF buffer at pH 6.5 was prepared from potassium dihydrogen phosphate and potassium chloride to obtain 0.03 M phosphate concentration and 0.13 M ionic strength.

Complex media preparation steps (Figure 1A): ethanolic stock solutions were prepared for each individual media component TCDC, TC, GDC, GC, CH, OA and PC. The desired amount of each component was weighed manually in 20 mL glass vials and dissolved in ethanol to obtain the desired concentration. The vials were then placed in a 2 × 4 plate (“Ethanolic Stock Solution” in Figure 1B) on the CM3 platform. A protocol was then initiated on the platform to transfer via positive displacement tips (PDT) aliquots from the ethanolic stock solution into 8 mL vials placed in a 4 × 6 plate (“Complex media” in Figure 1B) to obtain the desired ratio of the components in the final complex media. Ethanol was removed from the vial by placing the “Complex media” plate either in a GeneVac vacuum centrifuge at 40 °C until dry, or in a vacuum oven at 40 °C overnight. Finally, the “Complex media” plate was placed back on the CM3 platform, blank FaSSIF or FeSSIF buffers were added to the 8 mL vials via PDT and the vials were agitated (500 rpm, tumble stirring) to obtain clear solution or homogeneous emulsions to be used for the solubility study.

Biorelevant solubility experiment: 4–10 mg of API was dispensed in 4 mL vials (“Master Plate” in Figure 1B) using the Quantos QB5 platform and a stir bar was added to each vial. The “Master Plate” was then placed on the CM3 platform and a protocol was started: first 2 mL of complex media were dispensed via PDT in each vial of the “Master Plate”; each medium was dispensed in three vials to obtain results in triplicate. Then, the plate was heated to 37 °C and, once at temperature, the stirring (550 rpm, thumble stirring) and the timer started. At defined intervals (1, 2, 4 and 24 h), the protocol was paused, and the “Master Plate” was placed in the pre-heated (40 °C) centrifuge and centrifuged at 2000 rpm for 15 min. In total, 230 µL of supernatant solutions were sampled with a multichannel pipette and 230 µL of ethanol added as dilution solvent. Before resuming the protocol, the pH in each vial was measured with the CM3 platform (solid state pH probes).

HPLC analysis: quantitative HPLC analysis was carried out on an Agilent 1260 instrument using a Waters XSelect CSH C18 (30 × 2.1 mm, 2.5 μm) column. The column was heated at 60 °C and the following gradient was applied (Table 1), where mobile phase A is water + 0.05% TFA, and mobile phase B is acetonitrile + 0.05% TFA.

The samples were detected via DAD set at 220 nm. A calibration curve of GSK254 was developed, covering the full concentration range, and used for quantitation.

Table 2 shows the compositions used based on the contents of duodenal aspirates collected from healthy volunteers. In particular, composition 1 reflects the bile salts and main fatty acid (i.e., oleic acid, OA) contained in duodenal aspirates based on Moreno et al. [6] and composition 2 based on Riethorst et al. [7].

### 2.3. Permeability Measurements

Passive permeability measurements of GSK254 were conducted with MDCKII-MDR1 cells using working solutions at a concentration of 3 μM of GSK254, amprenavir, atenolol and propranolol in the presence of the potent P-gp inhibitor, GF120918.

The experiments were conducted using FaSSIF at pH 6.5 and pH 7.4 and FeSSIF at pH 5.8 and pH 7.4, both containing GF120918, as the transport medium in the apical compartment and DMEM (Dulbecco’s Modified Eagle’s medium) containing GF120918 and 1% Human Serum Albumin (HSA) as the transport medium in the basolateral compartment. The passive permeability was measured in one direction (apical-to-basolateral [A→B]) in triplicates.

The cell-line was incubated at 37 °C for a time period of 90 min prior to sampling. Integrity of the cell monolayer over the duration of the experiment was evaluated using paracellular permeability marker Lucifer yellow CH (100 μM).

### 2.4. Clinical Studies

Details for the clinical studies used for the development, qualification and verification of the model can be found elsewhere: FTIH-(ClinicalTrials.gov identifier, NCT03231943) [1], phase IIa study [2], food effect study (ClinicalTrials.gov identifier, NCT04263142) [11] and mass balance Clinical Absorption, Distribution, Metabolism and Excretion (ADME) study [12].

### 2.5. PBBM Modelling Strategy

The GSK254 PBPK model was built using the Simcyp Simulator V20TM (Certara UK Limited) and the food staggering module was selected to activate the dynamic pH, fluid volumes, bile salt secretion models [13] and particle population balance (PPB) model. Moreover, all simulations were carried out using the Healthy Volunteer population. Table 3 lists the input parameters used in the absorption and disposition model of GSK254. In cases where the input parameter is not measured or not relevant to the model application, default values were used.

#### 2.5.1. Dissolution and Absorption

The mechanistic ADAM model was used to handle dissolution using the Diffusion Layer Model (DLM), based on the Sugano model [14], whereas the mechanistic Permeability (MechPeff) model was used to provide regional permeability of GSK254 (details can be found in Pade et al. [15]).

Figure 2 shows the workflow followed to develop, qualify and verify the PBBM model with details outlined as follows:(1)GSK254 solubility in aqueous media was below the limit of quantification. Hence, it was not possible to measure a quantitative value for the intrinsic solubility (S_0_) of the drug. Therefore, a value of 0.00015 (mg/mL) was used after fitting FTIH clinical data, i.e., across a range of doses. However, the sensitivity analysis showed that any value below 0.001 (mg/mL) showed no significant difference in the predictions in both fasted and fed states.(2)A mass balance (ADME) study alongside the in vitro permeability data was used to set the parameters for the MechPeff model for the fed state.First, deconvolution of the plasma concentration profile of an immediate release (IR) tablet formulation (single dose—200 mg) was performed to get the dissolution profile that was directly imported to the ADAM model in order to remove at this development stage any issue with population variability in solubility/dissolution of the drug in fed and focus on capturing the fraction of the drug absorbed in the enterocytes (i.e., fa) and hence the pharmacokinetic (pk) data of the drug.Parameter Sensitivity Analysis (PSA) was performed to understand the interplay of reference concentration for permeation, ion permeation and LogK_M:W_ on the pharmacokinetic parameters of GSK254, such as Cmax and Tmax. PSA was performed at this development stage at fixed intrinsic transcellular permeability (P_trans,0_) predicted from octanol-to-water partition coefficient (LogP_O:W_). The reason for looking to the reference concentration for permeation is that the assumption behind the two options available in the MechPeff model (i.e., free fraction or total concentration) is that the free monomer can be absorbed by the enterocytes and secondly, the diffusion of the bound drug to micelles is lower compared to free monomer, which will affect the mucus permeability (P_UBL_) and hence the overall permeability. The decision of which option was chosen was made based on assessing the outcomes of the PSA with respect to the clinical data as well as with the observations from the in vitro permeability data in the presence of bile salt micelles. Similarly, the in vitro permeability data was used to decide if ion permeation was occurring or not.Another adjustment was to significantly reduce the absorption of GSK254 in the colon (see Table 3). GSK254, upon absorption, is metabolized to its glucuronide via UGT glucuronidation in the liver. This metabolite is secreted via biliary clearance and re-ejected to the small intestine. Because of its high hydrophilicity, this metabolite does not get absorbed and hence it is accumulated in the colon. Upon bacterial metabolism, the glucuronide is converted back to the parent drug. However, the absorption of the formed GSK254 is negligible and this is supported by the PO-ADME study in which the faeces losses were >94% in the parent drug [12]. Moreover, no GSK254-glucoronide was detected in the faeces, suggesting almost complete conversion of the glucuronide back to GSK254 in the gut with no significant re-absorption of GSK254. This can be further supported from the IV-ADME study where there were no secondary peaks, suggesting no enterohepatic recirculation of GSK254 after bacterial metabolism of its glucuronide form. Thus, reducing the absorption in the colonic compartment in the PBBM model is a valid approach supported by clinical data.(3)SIVA v4 was used to estimate LogK_M:W_ values from the in vitro solubility measurements and the values imported into the ADAM model.(4)PBBM model qualification was performed using a food effect relative bioavailability (rBA) study and FTIH study at selected doses (e.g., 200 mg, similar to the dose given in the ADME study).(5)PBBM model verification was performed using food effect rBA studies as well as FTIH and phase IIa proof-of-concept study at different multiple doses.

**Figure 2 pharmaceutics-15-00521-f002:**
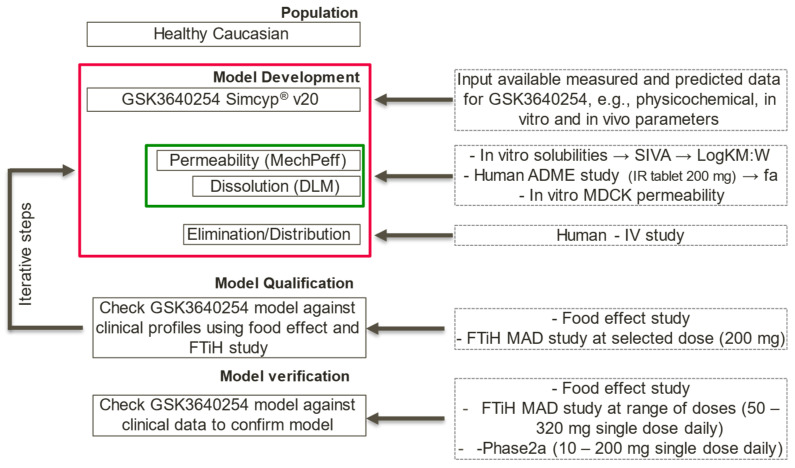
Workflow for PBPK model development, qualification and verification.

#### 2.5.2. Distribution and Elimination

The distribution models evaluated during model development used either minimal or full PBPK model approaches. Using the method of Rogers and Rowland and a full PBPK model within the Simcyp^®^ v20 simulator, a predicted volume of distribution (Vss) of 0.34 L/kg was correlated reasonably with the measured Vss of 0.36 L/kg from the balance/excretion and metabolism of [^14^C]-GSK254 following a single intravenous radio-labelled microtracer dose (concomitant with a non-radiolabeled oral dose) and a single oral radiolabeled dose (accepted manuscript). 

In humans, GSK254 has an in vivo average total plasma intravenous (IV) clearance of 1.04 L/h and renal clearance of 0.02 L/h (ADME study). The data from a bio-transformation study of [^14^C]-GSK254 after a single oral dose in humans indicated that GSK254 was extensively metabolized after being absorbed with less than <20% excreted as unchanged in bile. Therefore, the assigned biliary clearance was estimated according to the % of unchanged parent detected in the bile sample. In addition, from the same bile analysis, ~15% of GSK254 was converted to oxidative metabolites (fm,_CYP_ = 0.15) and ~45% of GSK254 was metabolized to acyl-glucuronidation (fm,_UGTs_ ~0.45). Furthermore, the results from an in vitro enzyme phenotyping suggested that CYP3A4 was the major contributing enzyme for the oxidative metabolites whereas UGT1A4 and UGT2B7 were the predominant enzymes for the direct acyl-glucuronidation metabolite. Using the Simcyp Retrograde Translation Tool and assigning 15% to CYP metabolism, the oxidative CLint was estimated (0.0027 μL/min/pmol) and listed as recombinant CYP3A4 enzyme clearance. The remaining intrinsic hepatic metabolic clearance was assigned to glucuronidation. The data from in vitro UGT reaction phenotyping studies with selective UGT1A4 and UGT2B7 inhibitor indicated that both enzymes have a similar % of contribution to acyl-glucuronidation formation (ADME study) (fm,_UGT1A4_ and fm,UGT2B7 ~0.45). Thus, a UGT1A4 CLint of 0.009 μL/min/pmol and a UGT2B7 CLint of 0.007 μL/min/pmol for glucuronidation metabolite with a biliary clearance of 0.111 μL/min/million hepatocyte cells for the intact GSK254 bile excretion were the input parameters for the overall GSK254 elimination model.

### 2.6. Prediction Accuracy and Statistical Analysis

The average absolute fold error (AAFE) was calculated as described by Shimizu et al. [16] to statistically compare the observed and simulated pharmacokinetic parameters as well as observed/predicted ratio. In addition, one-way ANOVA was applied to assess the statistical differences in the media composition (n = 3, *p* < 0.05).

## 3. Results

### 3.1. Impact of Biorelevant Media Composition and pH on the Equilibrium Solubility of GSK254

Considering the ampholytic nature of GSK254 and the observed positive food effect, the impact of media composition and pH on its solubility should be explored. This is needed to understand any source of in vivo inter-subject variability [2,11] that might be related to food-drug interactions, potentially affecting the solubility of GSK254 and hence its absorption.

An additional reason to look at the composition of the media is the difference be-tween a moderate- and high-fat meal observed in the food effect rBA study [11]. Although this difference was not clinically significant [11], the fact that the two meals do not affect the human gut physiology, in terms of gastric emptying, intestinal luminal pH as well as bile salt and phospholipids concentrations [17] led to the hypothesis that the presence of digestion products, such as fatty acids, might be the source of differences between the two meals. 

Examining the different components of the two meals, a high-fat meal contains more oleic acid whereas moderate-fat meals, in general, contain more saturated fatty acids with palmitic acid as the most abundant of this class. However, there is no available data for human intestinal aspirates collected after ingestion of a moderate-fat meal, such as the meals used in clinical studies of GSK254 [2,11] to determine if the palmitic acid would be the predominant digestion product in the lumen. Hence, the scope of this work, at this stage, was to develop an “average” PBBM model rather than a meal specific PBBM model. Hence, only the effect of oleic acid and cholesterol was assessed on the solubility of GSK254 at different pHs.

#### 3.1.1. Fasted

Understanding the factor, i.e., single vs. mixture of bile salts, with/without PC, OA and CH most affecting the solubility of GSK254 in fasted media at a fixed pH was the first step to assess at a high-level the drug–micelles interactions and the corresponding effect on the solubility of GSK254. Figure 3 shows the solubility of GSK254 in different compositions of fasted media at pH 6.5. The solubility results in fasted shows a single pH value. In an initial screening, using only taurocholate (TC), then TC and phospholipids (PC) and then TC+PC+cholesterol at different pH values showed no meaningful impact of pH on the solubility of the drug (in-house data). Thus, further exploration on the impact of the composition was conducted at a single pH value. The lowest solubility observed was either using a single bile salt (105 ± 16 μg/mL—Sol1: TC) or mixture of bile salts with different portions of TC, TCDC, GC and GDC (133 ± 15 μg/mL—Sol6_comp1 and 147 ± 33 μm/mL—Sol6_comp2). However, there was not a statistically significant difference between single and a mixture of BS as well as between the two compositions (*p* > 0.05).

The solubility of GSK254 was increased by 3.1 folds after PC addition (328 ± 14 μg/mL—Sol2: TC, PC vs. 105 ± 16 μg/mL—Sol1: TC). No statistically significant difference was observed between single (Sol2:TC, PC) and a mixture of BS (Sol7) as well as between the two compositions (Sol7: TC, TCDC, GC, GDC, PC _comp1 and 2), showing again that the BS composition does not impact the solubility of GSK254.

Next, CH was added to the media containing BS and PC (Sol3 and Sol8). The solubility of GSK254 was 317 ± 7.5, 342 ± 33 and 285 ± 37 (μg/mL) in Sol3, Sol8_comp1 and Sol8_comp2, respectively. No statistically significant difference between Sol3, Sol8_comp1 and Sol8_comp2 was observed. Moreover, the difference in solubility of GSK254 was not statistically significant between BS, PC, CH media (i.e., Sol3 and Sol8_comp1 and 2) and BS, PC media (i.e., Sol2 and Sol7_comp1 and 2), suggesting that CH did not alter GSK254 solubility.

Next, OA was added to BS, PC media with (Sol4 and Sol9_comp1 and 2) and without CH (Sol5 and Sol10_comp1 and 2). OA reduced GSK254 solubility by 1.5–2.5 folds depending on the media composition. In particular, the highest reduction was observed when single BS was used (Sol4—128 ± 4 μg/mL vs. Sol3—317 ± 7.5 μg/mL) followed by mixture of BS (175 ± 47 and 193 ± 47 μg/mL in Sol9_comp1 and 2, respectively vs. Sol8_comp1 and 2).

#### 3.1.2. Fed

Unlike the fasted media, oleic acid and cholesterol had a positive impact on the solubility of GSK254 at the same pH with fasted media, i.e., pH 6.5 (Figure 4). Interestingly, the solubility of GSK254 was shown to be higher in the typical FeSSIF media at pH 5 (Sol2: 1939 ± 67 μg/mL) compared to the media containing OA and CH, regardless of whether single (Sol4: 403 ± 48 μg/mL) or a mixture of BS (Sol9_comp1: 561 ± 58 μg/mL and Sol9_comp2: 599 ± 310 μg/mL) was added to the media. Furthermore, the results suggest that single and a mixture of BS will be affected in the same way as the solubility of GSK254 across all the pH values.

### 3.2. Effect of Biorelevant Media on Passive Permeation of GSK254 in MDCK Cells

MDCK cell line permeability experiments were used to obtain quantitative data for GSK254 passive permeability. Table 4 shows the results obtained using biorelevant media at different pHs. These experiments were performed to understand how pH and the presence of micelles both affect GSK254 passive permeability. The concentration in the donor compartment was fixed (2 μg/mL), which was well below the equilibrium solubility of GSK254 in FaSSIF media (328 μg/mL, see Sol2 in Figure 3). Thus, precipitation was not observed in biorelevant media as well as no solubility enhancement since elevated bile salt concentration in FeSSIF can be assumed, as there is no undissolved material in the donor compartment. The permeability of GSK254 was measured in the presence of a Pgp inhibitor.

The results showed the following:The pH affects GSK254 exact permeability (P_exact_) with the highest P_exact_ being observed at 7.4 (neutral state of the drug) in both media (i.e., FaSSIF and FeSSIF)The permeability of GSK254 was enhanced by the present of bile salts (see P_exact_ with DMEM solution, compared with biorelevant media)The permeability of GSK254 is bile salt concentration-dependent. This can be seen with the values of P_exact_ in FaSSIF and FeSSIF @pH 7.4 where P_exact_ is ~2.3 times higher in FeSSIFv2Ionized species of GSK254 do not contribute significantly to permeation. This is observed by comparing the P_exact_ values in FeSSIF_@pH7.4_ and FeSSIF_@pH5.8_. In particular, P_exact_ was three times lower in FeSSIF_@pH5.8_.Since there is no undissolved drug in the donor compartment, the enhanced permeability of GSK254 observed at elevated bile salt concentrations (i.e., in FeSSIF media) alongside the high affinity of the drug to micelles (see solubility measurements) suggests that the drug can potentially permeate whilst bound to micelles or the presence of mixed micelles alters the fluidity of the cell membrane, allowing this high molecular weight drug to permeate more effectively. This is a very crucial finding for the in vivo behaviour of GSK254, as the current assumption is that only the free monomers can be absorbed. Thus, in the fed state, the positive food effect might be caused by the bile salt mediated enhancement solubility and permeability. This last observation from the permeability data will be used to support the assumptions used in PBPK modelling (see Section 3.3).

### 3.3. PBBM Modelling

#### 3.3.1. Model Development

##### Distribution and Elimination

Elimination and distribution values were estimated by fitting the clinical systemic exposure of GSK254 after intravenous microdosing. The observed and simulated plasma GSK254 concentration profiles can be seen in Figure 5.

The observed vs. predicted ratio for mean AUC(0-t) and Cmax were 0.94 and 0.731, respectively. All of the PK parameters were within the two-fold error range (Table 5).

##### Dissolution and Permeability

GSK254 showed a positive food effect in vivo, attributed according to the in vitro data to high solubility and enhanced permeability in the presence of mixed micelles. A PBPK model was developed by integrating the in vitro data to further explore firstly if the in vitro data findings are clinically relevant, and secondly to explore other potential factors that might contribute to the positive food effect via PBPK modelling.

Since the exposure of GSK254 in the fed state should not be solubility-limited (Appendix A), the model development was initially focused on permeability aspects. Thus, the deconvoluted dissolution profile from the ADME study (IR 200 mg tablet) was used as a direct input to the model, to make sure that there were no issues with the dissolution, until the more mechanistic Diffusion Layer Model (DLM) handling dissolution was developed and applied.

PSA was performed to understand the interplay of reference concentration, ion permeation and LogK_M:W_ on pk parameters of GSK254.

Reference concentration is the concentration used to account for the concentration gradient promoting permeation. There are two options in the MechPeff model, either total concentration (free and bound drug to micelles) or free fraction of the drug. The assumption behind these two options is that first, only the free monomer can be absorbed by the enterocytes and second, the diffusion of the bound drug to micelles is lower compared to free monomer, which will affect the mucus permeability (P_UBL_) and hence the overall permeability [18]; a schematic explanation is given in Figure 6.

Thus, using the total concentration will enhance permeation because of the higher concentration gradient; the opposite holds using the free fraction. The bound and the free fraction of the drug will be affected by the LogK_M:W_ value, which will affect permeability as explained. It should be pointed out that the model accounts for the unionized (LogK_M:W-neutral_) and ionized species (LogK_M:W-ions_). However, LogK_M:W-ions_ is a single value applied to all ionized species, i.e., cations and anions. However, cations and anions might co-exist (Figure 7). Thus, the affinity of GSK254 into micelles may vary because of electrostatic repulsion and/or attraction forces between the zwitterionic GSK254 and the micelles based on the pH and the corresponding ionization status of the drug’s ionized species. For GSK254, even small fractions of cations and or ions presented can affect its solubility (Section 3.1). This will have a direct impact of the free fraction of the drug. As it can be seen from Figure 6, LogK_M:W_ is a global parameter applied in a luminal environment (total solubility) and a mucus microenvironment (mucus permeability, P_UBL_). However, the luminal pH in some individuals might be significantly different from the mucus that serves as a well-buffered microenvironment with narrower pH range or might be similar (Figure 7).

Despite this limitation, PSA was performed to understand how LogK_M:W_ impacts pk parameters of drugs such as Cmax and Tmax. PSA showed that only LogK_M:W-ionized_ had an impact on Cmax and Tmax (Appendix A). This is because a significant fraction of the drug is ionized for most of the physiological pH range and at neutral (zwitterion), close to pH 7.0–7.4. This can be seen in Figure 7 as well as from the HPLC analysis conducted to determine LogP, where the highest value was obtained at pH 7.4 (unpublished in-house data).

When the total concentration was used as ref. conc., LogK_M:W-ionized_ values > 5 should be used to predict Cmax close to the mean value observed in an in vivo food effect study (i.e., 1210 ng/mL). However, when only the free fraction was used, LogK_M:W-ionized_ had a negative impact on Cmax, as expected, as the higher the LogK_M:W-ionized_, the less free fraction will be available for absorption.

As explained in the in vitro permeability section, the bile salts promoted the permeation of the drug, even at lower free fraction because of the elevated bile salts concentration in FeSSIF media. Thus, total concentration was used in the model.

Furthermore, based on predicted ionization profile of GSK254 (Figure 7), ~10% of GSK254 is ionized at pH 5.8 compared to ~3% at 7.4. This difference is ~3.3 times, which is similar to the ×3 times difference in P_exact_ values of GSK254 observed between the two pHs in MDCK permeability experiments (Table 4). This shows that ion permeation is probably insignificant, and hence transcellular ion permeation was not allowed in the model.

SIVA v4^TM^, was used to estimate LogK_M:W_ values. Initially, all the experimental solubilities determined in the different biorelevant media were used to estimate the LogK_M:W_ values. As can be seen in Table 6, SIVA failed to capture the pH and compositional dependent solubility of GSK254 at the presence of mixed micelles. In particular, SIVA gave the same estimated solubility across the different pH values of the biorelevant media. 

In previously published works, SIVA was used to estimate the LogK_M:W_ values and predicted the experimental solubilities with high accuracy in biorelevant media of different versions across a range of compounds, such as ritonavir [21], ketoconazole [22,23], flurbiprofen [24], dipyridamole [23], itraconazole [23] and in a model ampholyte drug that behaves as a weak base within the physiological pH range in the GI tract [25]. In all these cases, the solubility was measured for each fasted and fed biorelevant media only at the typical average pH values of FaSSIF and FeSSIF media or with small difference between versions (e.g., Level II FaSSIF V1 (pH = 6.5) vs. Level II FaSSIF V3 (pH = 6.7)). Moreover, most of the compounds were either unionized in the small intestine (e.g., ritonavir [21]) or the ionization of the drug will be the same between the luminal and mucus pH. Thus, SIVA performance was assessed against cases where no pH-dependent drug-micelles interactions were involved.

Since SIVA cannot capture the pH- and compositional-dependent drug–micelle interactions, two sets of media were used to assess which version of FaSSIF and FeSSIF were most clinically relevant, i.e., adequately capturing the food effect. Therefore, it was decided the estimation of the LogK_M:W_ values to be performed separately using two sets of media. The first set is composed of two conventional biorelevant media (FaSSIF_@pH6.5_ and FeSSIF_@pH4.8_) and the second set of two biorelevant media containing CH and OA (FeSSIFCH+OA_@pH6.5_ and FeSSIFCH+OA_@pH4.8_). This approach was taken to ensure that the effect of CH and OA on the in-built linear bile salt concentration dependent estimation of the total solubility was properly accounted for. Table 7 shows the LogK_M:W_ values estimated using the two different sets of media. SIVA overpredicted the solubility of GSK254 in both versions of FaSSIF media but with good predictions in both FeSSIF media. The estimated LogK_M:W_ values were imported to the PBBM model and its predictivity was assessed.

GSK254 is practically insoluble in water and blank buffers so it was not possible to obtain an experimental value to inform the intrinsic transcellular permeability (Pt_rans,0_) of the unionized drug, which is an important parameter for the MechPeff model (see Equation (1) in Pade et al. [15]). In Simcyp^®^, there is an in-built correlation function to predicted P_trans,0_ based on the LogPo:w (so called octanol-to-water partition coefficient) value of the drug (Equation (1)). The model underpredicted the observed data in fed using Simcyp default values in the in-built function, regardless of using LogK_M:W_ values estimated either from the FaSSIF/FeSSIF set or from FaSSIF_+OA+CH_/FeSSIF_+OA+CH_ (Appendix A). Then, the in-built correlation function was updated using Sugano et al.’s [14] values, and the model performance was reassessed.
P_trans,0_ = a × P_O:W_ ^ b(1)
where a = 24.172 × 10^−6^ and b = 0.415 (Simcyp^®^ v.20 default values) or a = 2.36 × 10^−6^ and b = 1.1 (Sugano [14] values).

In this case, the model significantly overpredicted the observed data, regardless of using LogK_M:W_ values estimated either form the FaSSIF/FeSSIF set or from FaSSIF_+OA+CH_/FeSSIF_+OA+CH_ (Appendix A). Thus, the next step was to optimize P_trans,0_ to capture the observed data derived from the ADME study [12].

However, when the optimized P_trans,0_ was used to predict the FTIH-MAD-200 mg clinical study, the model overpredicted the last dose (Appendix A). Thus, applying learn and confirm, the best option was to use the P_trans,0_ predicted using the in-built correlation function provided by Simcyp.

After setting the permeability, the DLM model was used to handle the dissolution of the drug. The input parameters can be seen in Table 3. The PBBM model, using the DLM, provided fairly similar predictions (Appendix A) to the deconvoluted dissolution profile regardless to the LogK_M:W_ values set used from Table 7.

#### 3.3.2. Model Qualification

The PBBM model, using DLM, showed comparable results between the two sets of LogK_M:W_ values (Table 7), and performance verification was conducted against the food effect rBA study [11] to assess if the model can predict the positive food effect observed as well as the magnitude of this effect on the pk parameters. Thus, the simulations were run in fasted and fed states using first the LogK_M:W_ values estimated from FaSSIF/FeSSIF media and then from FaSSIF_+OA+CH_/FeSSIF_+OA+CH_.

Figure 8 shows the performance of the model against the food effect study of the GSK254-200 mg-tablet formulation [11]. The model overpredicted the observed plasma concentration profile in fasted (Figure 8-Fasted) but it captured the clinical data in fed (Figure 8-Fed) when the LogK_M:W_ values, derived from conventional biorelevant media, were used. Thus, although the model captured the positive food effect (FE), it did not predict the magnitude of FE. The reason is that the solubility of GSK254 might be overestimated in FaSSIF media, leading to higher LogK_M:W_ values resulting in increasing absorption. One the other hand, the negative effect of OA on the solubility of GSK254 observed in the new FaSSIF media led to lower LogK_M:W_ values, which allowed for properly capturing the clinical data in the fasted state. This difference in LogK_M:W_ values estimated does not affect the predictions in the fed state as does the permeability limited process. Thus, the total solubility is high enough using either set of LogK_M:W_ values to hit the upper limit for absorption, i.e., permeability. However, this is important for fasted state, as it is solubility limited, and hence proper estimation of the in vivo total solubility is required in order for the model to properly capture the magnitude of the FE observed. Based on these findings, the LogK_M:W_ values estimated using the biorelevant media containing OA and CH were used in the model for further verification.

The model was further qualified by performing simulation for multiple ascending doses of 200 mg—IR tablet of GSK254 given once per day for 14 days in healthy volunteers after a moderate-fat meal [2]. The model overpredicted the first dose but more importantly properly captured the steady state (after 6–7 days) and the final dose profile observed in the FTIH study (Figure 9). Based on these results, the model was qualified, as the drug clinical efficacy is achieved at a steady state and hence it is important that the model captures it, as it was used for further verification.

#### 3.3.3. Model Verification

The qualified model was verified by assessing its performance against clinical data from the FTIH study, where GSK254 was administrated in multiple ascending doses of different strengths [1], and from another food effect study where 150 mg (2 × 25 mg and 1 × 100 mg)-IR-tablet of GSK254 was given to healthy volunteers after a moderate-fat meal. The qualified PBPK model was able to recover the plasma concentration profile of GSK254 in fed healthy volunteers across all doses (Figure 10).

The model was also verified against the clinical data of the phase IIa study [2]. The PBBM model was able to capture the plasma concentration profile of GSK254 in fed healthy volunteers for most of the doses administrated (Figure 11).

Table 8 summarizes the predicted and observed pk data across all the clinical studies used to develop, qualify and verify the model.

Figure 12 shows the overall predictiveness of the model by taking the ratio of observed to predicted pk values. All the ratios were within two-fold, with most of them within 1.25-fold.

Moreover, the AAFE measures for Cmax, Tmax and AUC_0-t_ for observed and simulated pharmacokinetic parameters were found to be 1.23, 1.15 and 1.30, respectively (Table 8).

## 4. Discussion

In the presented work, a PBBM model was developed to predict the PK of GSK254 after IV and oral administration. The presented model successfully simulated both IV and oral PK profiles of GSK254.

PBBM modelling has attracted the attention of both the pharmaceutical industry and regulatory agencies [26,27] as an emerging technology to predict clinical food effects. The reason is that administration of drugs with food alters the human physiology in many ways including increased gastric, intestinal and biliary secretions, prolonged gastric emptying, changes in gastrointestinal pH values and increased splanchnic blood flow with a direct impact on a drug’s solubility and absorption, potentially affecting its clinical safety and efficacy. Thus, food–drug interaction is a multi-factorial process whereby is not easy to assess the impact of each component separately and/or any of their synergistic effect via in vivo and/or in vitro dedicated studies. On the other hand, modelling can handle many of these factors simultaneously, allowing via PSA or global sensitivity analysis (GSA) to potentially understand what the main reason of the food effect is and make a link between formulation/drug characteristics and in vivo performance. It is important to recognize that PSA or GSA can only serve as indicators of the potential cause of the food effect, and further in vivo and in vitro studies are required to mechanistically understand the underlying mechanism of the food effect.

This is because there are still limitations and gaps in the PBPK platforms to mechanistically describe food–drug interactions. One of the limitations, addressed in this study, is the pH and compositional dependent interactions of different ionized species co-existing with mixed micelles and how these are changing, if changing, between the luminal and the mucus microenvironment, especially for zwitterionic molecules. Another gap is the regional, time variant qualitative/quantitative changes in the bile salts, mono-glycerides, free fatty acid and phospholipids upon selective absorption of the endogenous and food digestion components along the GI tract. For instance, passive absorption of glycine-conjugated and free chenodeoxycholic bile salts occurs mainly in jejunum whereas taurine-conjugated and free bile acids (e.g., cholic acid) are actively absorbed in the ileum [28]. These changes will alter the pH- and non-pH-dependent drug–micelles interactions. Thus, there is a need to better model these regional differences not only by understanding their impact on the solubility of the drug but also on its permeability. Although the dynamic model of the total bile salt regional profile is provided in a Simcyp^®^ v20 simulator with inter- and intra-subject variability [13,29], the lack of regional time variant composition of mixed micelles and the corresponding pH- and non-pH dependent drug–micelles interactions limits the ability to mechanistically understand these phenomena. For instance, it was challenging to develop a meal-specific PBBM model to understand the lower Cmax and AUC observed with a high-fat meal compared to a moderate-fat meal. However, the challenge was not only attributed to the current complexity of the Simcyp platform (e.g., lack of regional LogK_M:W_), but also to the lack of in vivo data providing information about the luminal composition upon digestion of a typical FDA moderate-fat meal as well as lack of appropriate biorelevant media. Looking to the overall composition of the two meals, the high-fat meal generally contains more unsaturated fatty acids with oleic acid as the most predominant. On the contrary, the moderate-fat meal contains more saturated fatty acids with palmitic acid as the most predominant of this class. However, it is not known if palmitic acid is indeed the main free fatty acid presented in the luminal contents, but which should be accounted for in biorelevant media to assess any potential alternations in the solubility and permeability of lipophilic drugs and inform the model. 

Differences in the viscosity of the two meals impacting dissolution and/or gastric emptying could be another reason to explain changes observed in the pk parameters of GSK254 between the two meals. However, this seemed to be unlikely as the viscosity should impact gastric emptying [30], but based on the available in vivo data, the gastric emptying was similar between the two meals [17]. It should be noted that the two meals used in Armand et al. [17] were a homogenized Ensure plus-based formula compared to the normal non-homogenized meals used in the clinical studies in this work. Looking to TIM-1 experiments (see Appendix A for details about protocols followed), the bioaccessibility of the drug was reduced ~20% with a high-fat homogenized meal compared to the moderate-fat meal (Appendix A). Again, homogenized meals were used, and the gastric emptying was the same for both meals (half-life time set to 80 min). Although, the impact of viscosity on the gastric emptying was excluded from the TIM-1 experiments, the bioaccessibility of the drug was reduced by the same% as the Cmax in the clinical studies (~21–24%). These results point to the potential differences in the luminal composition upon digestion of the two different meals. Considering the sensitivity of the solubility of GSK254 in the media composition and pH, the potential reason behind the reduced Cmax and AUC in the high-fat meal might be caused by changes in the composition of mixed micelles and the corresponding interactions with an ampholyte molecule such as GSK254. Solubility data showed that it was the presence of fatty acids and cholesterol and not the composition in the bile salt mixture affecting the solubility of GSK254.

Another unknown factor is how the luminal composition affects the passive permeability of GSK254, as it is a relatively high molecular weight drug. The presence of different bile salts, phospholipids and free fatty acids upon digestion of different fat-containing meals might differently affect the fluidity of the epithelium membrane, resulting in alternations in the passive permeability of a high molecular weight drug. It is also not clear how the presence of different colloidal structures formed upon digestion of different fat-containing meals affect GSK254 permeability. As mentioned above, the presence of a higher concentration of bile salts and phospholipids in the biorelevant media (i.e., FeSSIF) enhanced the permeability of GSK254. However, the reason/mechanism underlying this enhanced passive permeability is not well understood. Thus, the only available option in the PBBM model to account for this enhanced permeability is to allow the free and the bound fraction to permeate the enterocytes.

Another challenge was to understand the role of P-gp and BCRP in GSK254 disposition and absorption. The results from single and multiple escalation dose study [1] indicated a trend towards a less than dose-proportional increase at higher doses. This phenomenal can be explained by either the saturation of drug metabolism or drug absorption [31]. The role of efflux transporters such as P-gp and BCRP had been proven to limit drug intestinal absorption [32]. In addition, owing to the poor recovery observed in the in vitro efflux transporters, the results of in vitro studies to determine the role of P-gp and BCRP in GSK254 disposition were inconclusive (unpublished inhouse data). Therefore, the current model version did not incorporate this pathway to describe the overall GSK254 intestinal absorption. This could potentially impact the prediction of GSK254 exposure at a dose level less than 100 mg observed for single and multiple dosing regimens (Figure 11).

Regardless of these limitations/challenges, an “average” (i.e., a non-meal specific) PBBM model was developed and successfully described the in vivo data for both meals. This was possible by systematically investigating the impact of media composition and pH on the solubility of GSK254, which allowed for estimating these model parameters, in particular LogK_M:W_, which impacts both dissolution and permeability.

## 5. Conclusions

A PBBM model was successfully developed and applied to predict the food effect of a BCS class IV zwitterionic drug with complex interactions with mixed micelles. In vitro data alongside PBBM modelling suggested that the positive food effect observed in the clinical studies was attributed to micelle-mediated enhanced solubility and permeability. This work showed that the predictive power of PBBM is improved when the understanding of the food effect goes beyond the typical approach (e.g., simply use of the typical FaSSIF and FeSSIF media) as well as when high quality in vitro data is integrated. Furthermore, although the integration of all the available in vitro data might not be always possible or the model might not include all the underlying mechanisms, the in vitro data should be used (if possible) to inform the right assumptions and adjustments to the model to describe the in vivo data. For instance, if TIM-1 data was lacking, then the potential explanation about the differences between the two meals observed in the exposure of GSK254, could be based on variability in the gastric emptying rather than differences in food composition (e.g., fatty acids). Possibly a PSA might show this impact, however, that would lead to wrong conclusions. Thus, the need for exploring in future work how the composition might alter the absorption and hence the exposure of this class of drug is emphasised. Another key lesson from this work was that the development of the PBBM model should be conducted in collaboration with the people who generate the in vitro and in vivo data, in order to have a common understanding of the data generated. The developed model strategy can be effectively adopted to increase the confidence of using PBBM models to predict food effect of BCS class IV drugs and build on this foundation for defining a safe space and potential clinically relevant dissolution specifications.

## Figures and Tables

**Figure 1 pharmaceutics-15-00521-f001:**
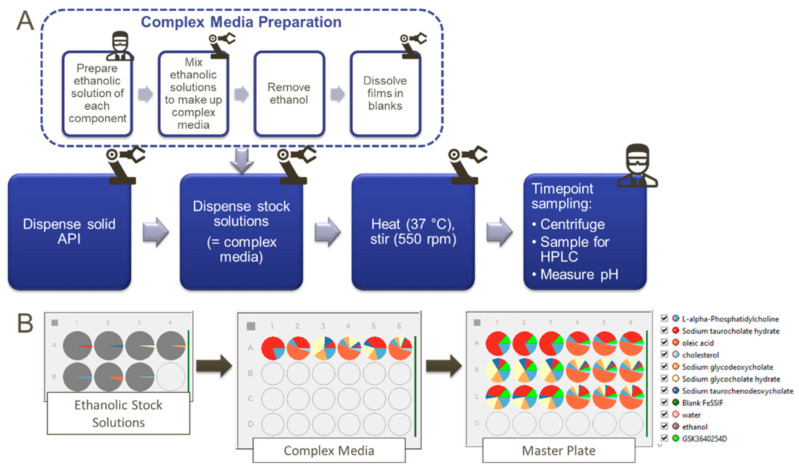
(**A**) Workflows schematic for preparation of complex biorelevant media (**top**, dashed box) and for biorelevant solubility experiment (**bottom**). The icons on top of each box indicate if the step is run manually (scientist icon) or on an automated platform (robotic arm icon); (**B**) Example of plate designs highlighting the ratio of the various biorelevant media components and API as pie-charts for each plate position. NOTE: blank FeSSIF buffer is not shown for easier view of media composition.

**Figure 3 pharmaceutics-15-00521-f003:**
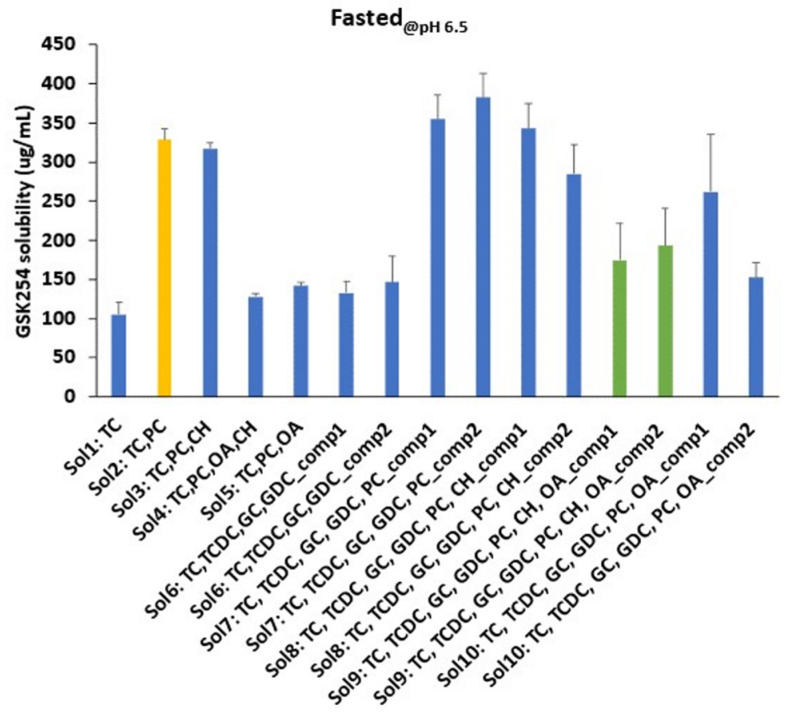
Bar chart (mean, ±SD) of GSK254 equilibrium solubility (24 h) values in different compositions of fasted media at pH 6.5. sodium taurochenodeoxycholate (TCDC); sodium taurocholate hydrate (TC); sodium glycodeoxycholate (GDC); sodium glycocholate hydrate (GC); cholesterol (CH); oleic acid (OA); L-alpha- phosphatidylcholine from egg yolk (PC). Typical FaSSIF media composition (yellow bar). Composition of the new media containing mixture of bile salts as well as OA and CH showing the negative impact of OA on GSK254 solubility (green bars). Refer to Table 2 for the exact recipe of composition 1 (comp1) and composition 2 (comp2); solubility measurements were conducted in each composition in triplicates (n = 3).

**Figure 4 pharmaceutics-15-00521-f004:**
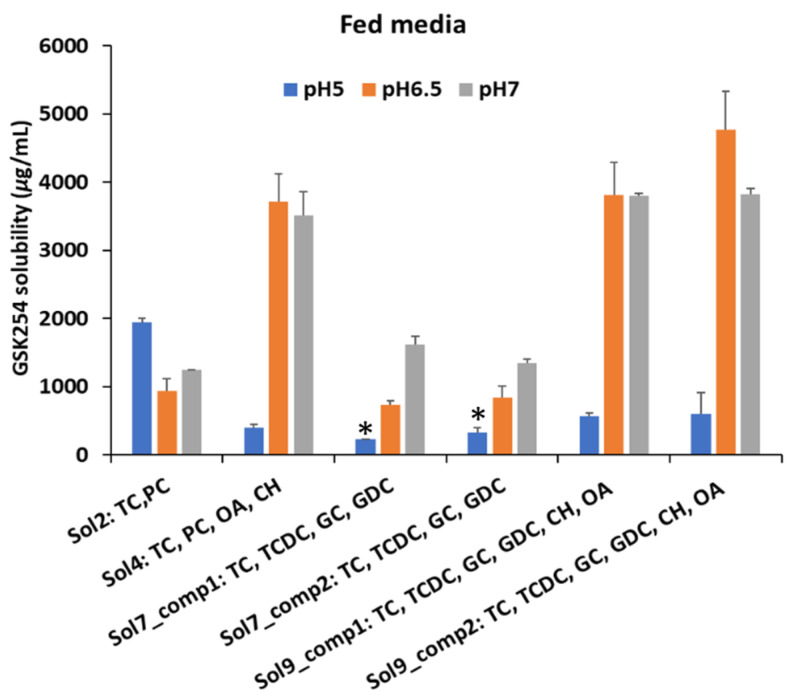
Bar chart (mean, ±SD) of GSK254 equilibrium solubility (24 h) values in different compositions of fed media at 5, 6.5 and 7 pH; sodium taurochenodeoxycholate (TCDC); sodium taurocholate hydrate (TC); sodium glycodeoxycholate (GDC); sodium glycocholate hydrate (GC); cholesterol (CH); oleic acid (OA); L-alpha- phosphatidylcholine from egg yolk (PC). Refer to Table 2 for the exact recipe of composition 1 (comp1) and composition 2 (comp2). * Sol7_comp1&2@pH5 do not contain PC; solubility measurements were conducted in each composition in triplicates (n = 3).

**Figure 5 pharmaceutics-15-00521-f005:**
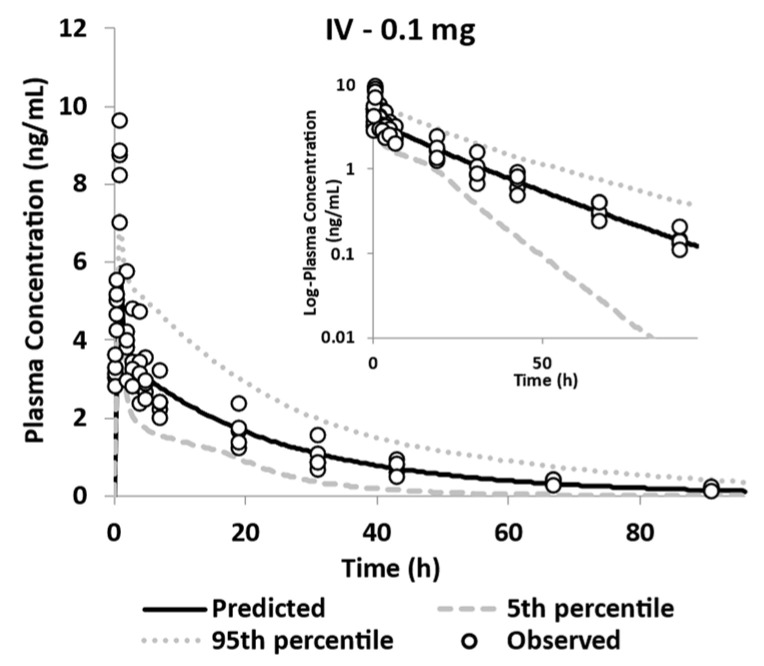
Observed and simulated plasma GSK254 concentration profiles of IV microdosing of GSK254 in healthy volunteers (n = 5, ADME study [12]).

**Figure 6 pharmaceutics-15-00521-f006:**
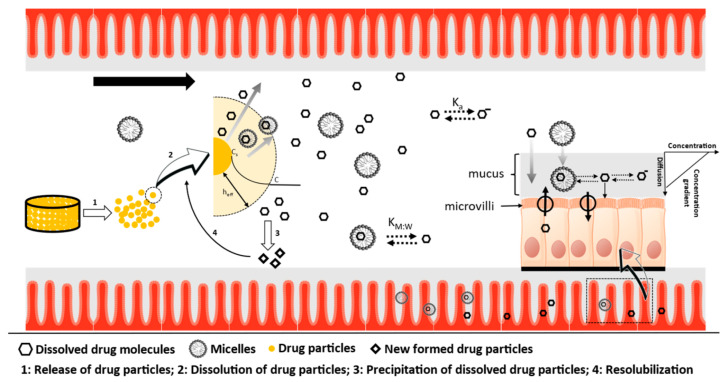
Schematic representation of the phenomena that take place in the luminal environment of the human GI tract affecting the performance of the dosage form and the absorption of the drug molecules; Ka ionization constant; K_M:W_: micelle-to-water partition coefficient; Cs: drug concentration at the surface of the drug particle; C: bulk concentration of the drug.

**Figure 7 pharmaceutics-15-00521-f007:**
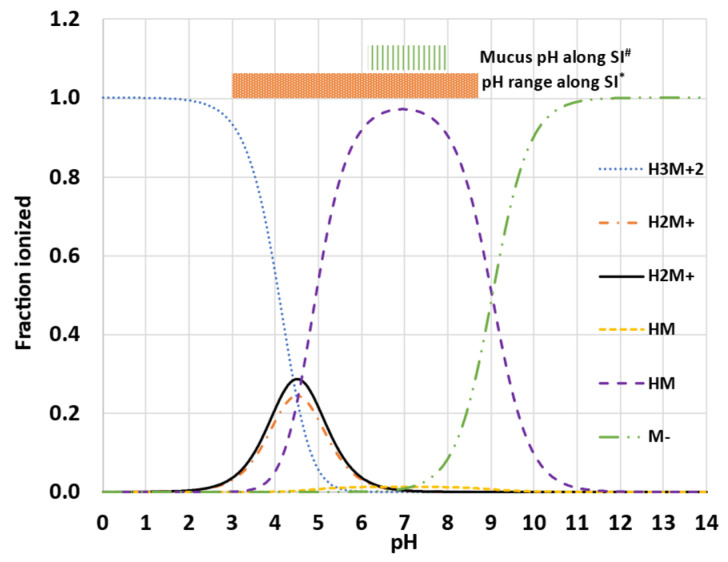
Predicted ionization profile of GSK254 using ADMET predictor v.10.0 (Simulation Plus, USA). The inserted horizontal bars show the range of the pH along the small intestine (SI) and the mucus. * Range of luminal pH in the SI observed in human aspirates at different regions of SI (duodenum [6,7], jejunum [7] and ileum [19]); # Mucus pH along the SI measured in porcine gastrointestinal mucus [20]. The zwitterion is indicated by the dashed line (HM).

**Figure 8 pharmaceutics-15-00521-f008:**
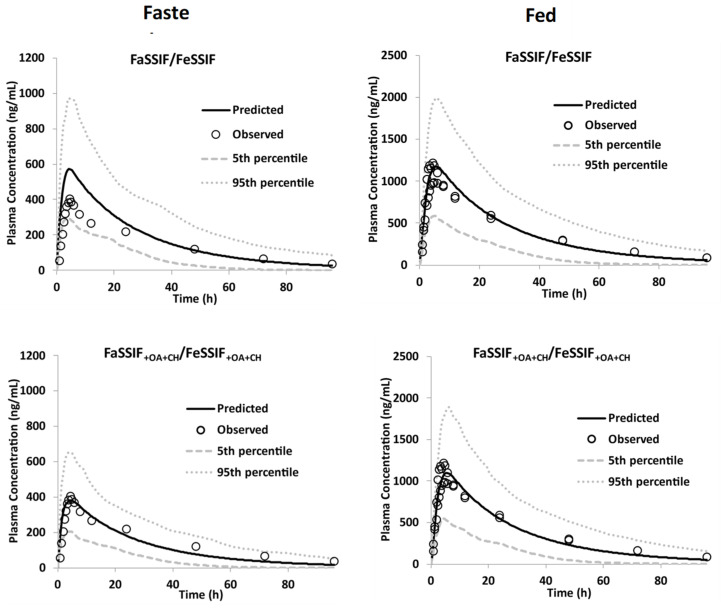
Performance verification (PV) of the PBBM model in fasted and fed states using the DLM model; overlay observed data from food effect study—200 mg (2 × 100 mg)—IR tablet—SD—after moderate fat [11]; PV was performed separately using LogK_M:W_ values estimated from the FaSSIF/FeSSIF media and from the FaSSIF_+OA+CH_/FeSSIF_+OA+CH_ media.

**Figure 9 pharmaceutics-15-00521-f009:**
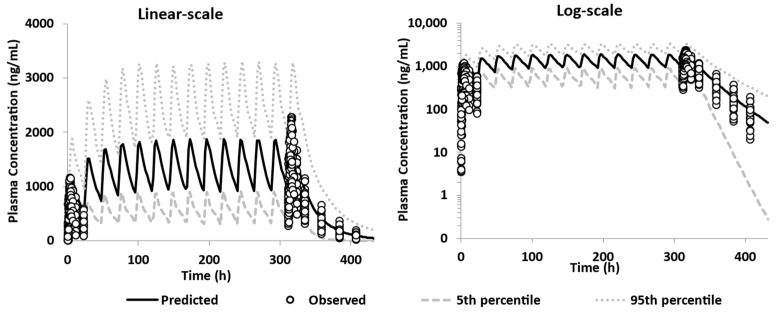
Performance verification of the PBBM model in fed state using the DLM model; overlay observed data from FTIH MAD—200 mg—tablet—single dose daily after a moderate–high fat meal [2].

**Figure 10 pharmaceutics-15-00521-f010:**
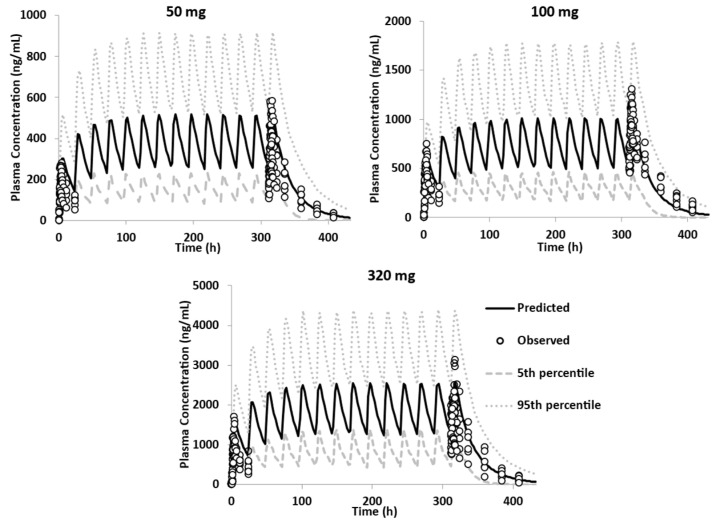
Performance verification of the PBPK model in fed state using the DLM model; overlay observed individual data from FTIH MAD–50, 100 and 320 mg—tablet—single dose daily after a moderate–high fat meal [1].

**Figure 11 pharmaceutics-15-00521-f011:**
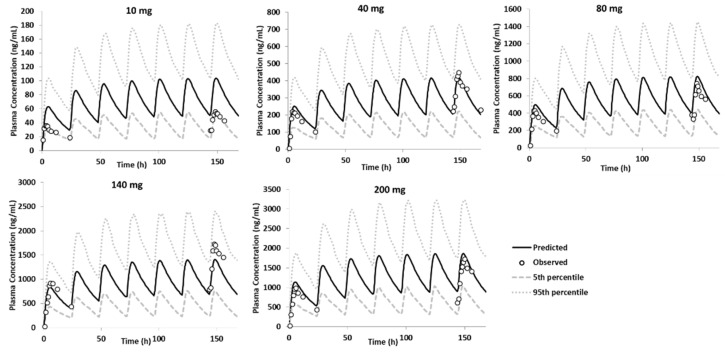
Performance verification of the PBPK model in fed state using the DLM model; overlay observed individual data from phase IIa study—10, 40, 80, 140 and 200 mg—IR tablet—single dose daily after a moderate–high fat meal [2].

**Figure 12 pharmaceutics-15-00521-f012:**
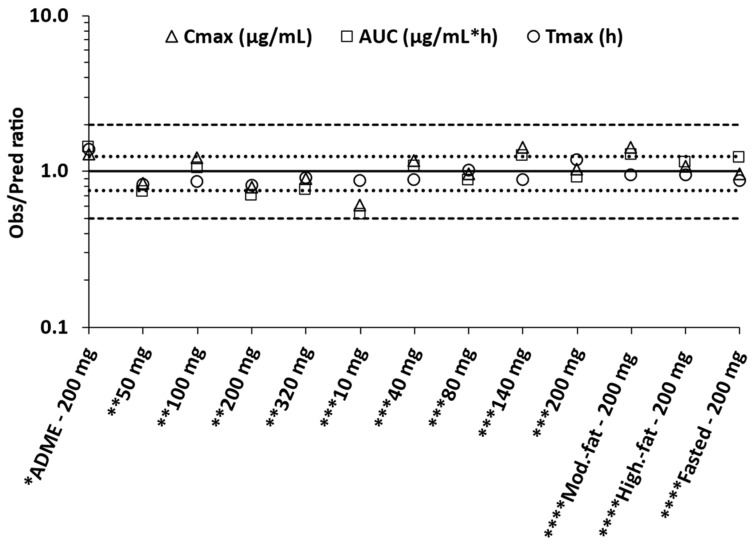
Observed versus predicted ratio of pharmacokinetic parameters of GSK254. The solid line represents the line of unity, dot lines 1.25 and dashed lines the 2-fold error thresholds. * Wen et al. [12], ** Joshi et al. [1], *** Spinner et al. [2], **** Johnson et al. [11].

**Table 1 pharmaceutics-15-00521-t001:** HPLC mobile gradient profile.

Time (min)	Mobile Phase A (%)	Mobile Phase B (%)	Flow Rate (mL/min)
0.00	75	25	1.5
1.00	20	80	1.0
1.01	75	25	1.0
2.00	75	25	1.0

**Table 2 pharmaceutics-15-00521-t002:** Composition of FaSSIF and FeSSIF media used to measure GSK254 equilibrium solubility (24 h).

Solution	Total BS (mM)		%				PC (mM)	OA (mg/mL)	CH (mg/mL)
		Composition	TC	TCDC	GC	GDC			
1	3 ^a^ 15 ^b^		100	-	-	-	-	-	-
2			100	-	-	-	0.75 ^a^, 3.75 ^b^	-	-
3			100	-	-	-	0.75 ^a^, 3.75 ^b^	-	0.027 ^a^, 0.72 ^b^
4			100	-	-	-	0.75 ^a^, 3.75 ^b^	0.53 ^a^, 6.5 ^b^	0.027 ^a^, 0.72 ^b^
5			100	-	-	-	0.75 ^a^, 3.75 ^b^	0.53 ^a^, 6.5 ^b^	-
6		1	14.9	15.1	45.1	24.9	-	-	-
		2	47.1	8.8	24.6	19.5	-	-	-
7		1	14.9	15.1	45.1	24.9	0.75 ^a^, 3.75 ^b^	-	-
		2	47.1	8.8	24.6	19.5	0.75 ^a^, 3.75 ^b^	-	-
8		1	14.9	15.1	45.1	24.9	0.75 ^a^, 3.75 ^b^	-	0.027 ^a^, 0.72 ^b^
		2	47.1	8.8	24.6	19.5	0.75 ^a^, 3.75 ^b^	-	0.027 ^a^, 0.72 ^b^
9		1	14.9	15.1	45.1	24.9	0.75 ^a^, 3.75 ^b^	0.53 ^a^, 6.5 ^b^	0.027 ^a^, 0.72 ^b^
		2	47.1	8.8	24.6	19.5	0.75 ^a^, 3.75 ^b^	0.53 ^a^, 6.5 ^b^	0.027 ^a^, 0.72 ^b^
10		1	14.9	15.1	45.1	24.9	0.75 ^a^, 3.75 ^b^	0.53 ^a^, 6.5 ^b^	-
		2	47.1	8.8	24.6	19.5	0.75 ^a^, 3.75 ^b^	0.53 ^a^, 6.5 ^b^	-

Note: Total concentration of different components in Fasted ^a^ and Fed ^b^ media; Composition 1 was developed based on Riethorst et al. [7] study whereas composition 2 was developed based on Moreno et al. [6] using mean values; bile slats (BS).

**Table 3 pharmaceutics-15-00521-t003:** GSK254 PBBM model input parameters.

Parameters	Value	Comments/Reference
**PhysChem and Blood Binding**		
Mol Weight (g/mol)	727.07	
Log P	5.49	Measured unpublished data
pKa 1	4.63	
pKa 2	8.65	
B/P	0.53	
fu	0.14	
**Absorption**		
Absorption Model	ADAM	
UBL fluid volumes	Yes	
fu(Gut)	0.226	Based on ADME study
Peff,man Type	Regional	
Permeability Method	Mechanistic Model	
Effective Concentration for Permeation	Total Concentration	Based on MDCK Papp data using FaSSIF/ FeSSIF media
MechPeff, P_trans,0_ (10^−6^ cm/s)	4565.23	Predicted by Simcyp
Include Ion Transcellular Permeation	No	Based on MDCK Papp data
Apply Accessible Surface Area Scalar	Yes	Default
Force Unstirred boundary layer pH to bulk pH	No	Default
Paracellular Scalar	0.00039	Default
P_eff,man_ (10^−4^ cm/s) _duodenum_	0.13	Predicted *
P_eff,man_ (10^−4^ cm/s) _jejunum I_	0.14 0.095	
P_eff,man_ (10^−4^ cm/s) _jejunum II_		
P_eff,man_ (10^−4^ cm/s) _ileum I_	0.037	
P_eff,man_ (10^−4^ cm/s) _ileum II_	0.037	
P_eff,man_ (10^−4^ cm/s) _ileum III_	0.037	
P_eff,man_ (10^−4^ cm/s) _ileum IV_	0.035	
P_eff,man_ (10^−4^ cm/s) _colon_	0.015	
Colon Abs scalar	0.001	GSK254 bioavailability from the colon is negligible based on ADME clinical data [12]
Input Form	Solid Formulation	
DLM Particle Handling Model	Particle Population Balance	
Formulation	IR: DLM Model	
Define Disintegration Profile	Not activated	
Dissolution Type	Solubility	
Solubility Type	Intrinsic (user)	
Solubility (mg/mL)	0.00015	Fitted using FTIH clinical data
Supersaturation Precipitation Model	First Order	
FO Precipitation Model	Model 2	
PRC (Precipitation Rate Constant)	Global	
PRC (1/h)	0.0001	
CSR (Critical Supersaturation Ratio)	Global	
CSR value	1000.00	
Reference Concentration for Precipitation Model	Unbound-Unionised	
Dispersion Type	Polydispersed	
Particle size distribution	Log Normal	Fitting measured data
Input Type	Volume Fraction	
D10 (µm)	3.2	Measured
D50 (μm)	8.1	
D90 (μm)	23.1	
No. of Bins (Simulation)	50	Default
Particle density (g/mL)	1.2	Measured
Heff method selected	Hintz-Johnson	Default
Heff cut-off value (µm)	30	Default
Bile Micelle mediated solubilisation	On	
LogK_m:w,unionized_	0.019	Estimated with SIVA 4
LogK_m:w,ionized_	4.559	Estimated with SIVA 4
Segregated transit time model	Activated	
MRT of particles and pellets may be shorter than fluid	Activated	
Mucus regional thickness		Default values
**Distribution related parameters**		
Distribution Model	Full PBPK Model	
Vss (L/Kg)	0.34	Predicted
Prediction Method	Method 2	
Kp Scalar	1.0	
**Elimination related parameters**		
Clearance Type	Enzyme Kinetics	Determined based on ADME study [12]
In vitro metabolic system	Recombinant	
Enzyme	CYP3A4	
CLint (µL/min/pmol)	0.003	
fu mic	1.0	
Enzyme	UGT1A4	
CLint (µL/min/pmol)	0.009	
fu mic	1.0	
Enzyme	UGT2B7	
CLint (µL/min/pmol)	0.007	
fu mic	1.000	
ISEF	1.000	
Additional HLM CLint (μL/min/mg protein)	0.354	
Biliary CLint (Hep) (µL/min/10^6^)	0.111	
CL R (L/h)	0.020	

Note: * The population representative regional Peff,man (i.e., average subject) is taken into account by the model during simulations. The values reported in the table do not include the effect of bile micelle binding on effective permeability. This is considered during simulations to account for population variability.

**Table 4 pharmaceutics-15-00521-t004:** Apparent exact permeability of GSK254 in biorelevant media using MDCK cell lines.

Compound	Drug (µM)	Rate A > B (nmoles/cm^2^/h)	s.d.	A > B M.B (%)	s.d.	P_exact_ (nm/s) A > B	s.d.
GSK254 + GF120918 (DMEM/DMEM)	2.67	0.00	0.0	74	8.3	0.0	0.0
GSK254 + GF120918 (FaSSIF pH7.4)	2.67	0.031	NA	109	NA	29	NA
GSK254 + GF120918 (FaSSIF pH6.5)	2.67	0.015	0.0041	97	2.4	16	4.5
GSK254 + GF120918 (FeSSIF pH7.4)	2.67	0.055	NA	87	NA	67	NA
GSK254 + GF120918 (FeSSIF pH5.8)	2.67	0.021	0.0068	96	9.7	23	6.9

Note: Apical to Basolateral (A > B); the data is presented in mean and standard deviation (s.d); %Mass Balance (%M.B). Permeability experiments were conducted in triplicates in each media (n = 3).

**Table 5 pharmaceutics-15-00521-t005:** Observed vs. predicted pk parameters of IV microdosing of GSK254 in healthy volunteers (ADME study [12]).

Geometric Mean (%CVb)-0.1 mg [^14^C]-GSK254 IV Dose (n = 5)					
	C_max_ (ng/mL)	AUC_(0-t)_ (ng/mL·h)	Vss (L/kg)	CL (L/h)	T_1/2_ (h)
Simulated	5.98 (16)	87.8 (41)	0.34 (17)	1.14 (41)	17.37 (37)
Observed	8.44 (12.0)	93.3 (20.4)	0.36 (0.27)	1.04 (19.7)	21.7 (12.6)
Pred/Obs	0.71	0.94	0.94	1.10	0.80

Note: Maximum observed concentration (Cmax); area under the curve (Area under the plasma drug concentration-time curve from pre-dose to the end of the dosing interval at steady state)—(AUC_(0-t)_); volume of distribution at steady state (Vss); clearance (CL); apparent terminal phase half-life (T_1/2_); % between-participant coefficient of variation (%CVb).

**Table 6 pharmaceutics-15-00521-t006:** Experimental vs. SIVA v4 predicted solubilities of GSK254 and different biorelevant media.

Medium	pH	Solubility (mg/mL)	
		Experimental	Predicted
FeSSIF_+CH+OA_	6.9	3.71	2.03
	6.2	4.1	2.03
	5	0.52	2.03
FeSSIF	6.99	1.24	2.03
	6.2	0.94	2.03
	4.8	1.94	2.03
FaSSIF	6.5	0.36	0.71
FaSSIF_+CH+OA_	6.5	0.17	0.71

**Table 7 pharmaceutics-15-00521-t007:** LogK_M:W_ estimated values using SIVA v4 predicted solubilities of GSK254 and different sets of biorelevant media.

	LogK_M:W_	Media Set	Solubility (mg/mL)		
			Observed	Predicted	Pred/Obs
Neutral	1.05 × 10^−6^	FaSSIF_@pH6.5_	0.36	0.65	1.81
Ion	5.149	FeSSIF_@pH4.8_	1.94	1.84	0.95
Neutral	0.019	FaSSIF_+CH+OA@pH6.5_	0.17	0.2	1.18
Ion	4.559	FeSSIF_+CH+OA@pH4.8_	0.52	0.51	0.98

**Table 8 pharmaceutics-15-00521-t008:** Simulated and Observed Pharmacokinetic Parameters of GSK254 following administration of an IR-tablet at different doses in fasted and fed conditions.

Study	Dose (mg)	State	Cmax (μg/mL) *		Tmax (h) **		AUC (μg/mL h) *	
			Obs.	Pred.	Obs.	Pred.	Obs.	Pred.
ADME-study [12]	200	^a^ Fed	1.3 (7.6)	1.0 (40)	7 (4.5–9.0)	5.0 (3.1–7.9)	41.2 (14.5) ^b^	28.7 (44)
FTIH [1] ***	50		0.41 (31.6) (95%CI: 0.30–0.57)	0.49 (43) (95%CI: 0.44–0.54)	3.8 (2.5–5)	4.57(2.45–7.25)	6.3 (34.0) (95%CI: 4.4–8.9) ^c^	8.4 (49) (95%CI: 7.4–9.4)
	100		1.18 (10.3) (95%CI: 1.1–1.3)	0.96 (43) (95%CI: 0.9–1.1)	4.0 (1.5–4.5)	4.60 (2.4–7.25)	17.5 (13.8) (95%CI: 15.2–20.2)^c^	16.5 (48) (95%CI: 14.6–18.5)
	200		1.40 (30.8) (95%CI: 1.2–1.6)	1.77 (42) (95%CI: 1.6–1.96)	3.8 (1.5–5.5)	4.62 (2.5–6.9)	21.5 (34.2) (95%CI: 18.7–24.7) ^c^	30.5 (47) (95%CI: 27.1–34.2)
	320		2.16 (20.4) (95%CI: 1.7–2.7)	2.4 (42) (95%CI: 2.2–2.7)	4.3 (1.5–6.0)	4.70 (2.6–6.7)	32.0 (35.4) (95%CI: 22.3–45.8) ^c^	41.7 (47) (95%CI: 37.1–46.8)
Phase IIa [2] ****	10		0.06 (41.3) (95%CI: 0.04–0.08)	0.098 (41) (95%CI: 0.089–0.11)	4.02 (1.87–5.00)	4.55 (2.45–7.3)	0.91 (44.7) (95%CI: 0.58–1.42)	1.7(47) (95%CI: 1.5–1.9)
	40		0.47 (20.6) (95%CI: 0.38–0.58)	0.4 (42) (95%CI: 0.35–0.44)	4.06 (2.00–8.00)	4.55 (2.4–7.25)	7.46 (26.8) (95%CI: 5.66–9.84)	6.8 (47) (95%CI: 6.1–7.6)
	80		0.75 (23.7) (95%CI: 0.59–0.96)	0.78 (41) (95%CI: 0.7–0.9)	4.58 (4.00–5.18)	4.5 (2.5–7.25)	11.80 (26.7) (95%CI: 8.98–15.6)	13.4 (46) (95%CI: 12–15)
	140		1.86 (26.0) (95%CI: 1.42–2.43)	1.3 (41) (95%CI: 1.2–1.5)	4.08 (2.92–5.2)	4.60 (2.55–7.0)	29.3 (27.9) (95%CI: 22.0–39.0)	23 (46) (95%CI: 20.5–25.8)
	200		1.86 (19.5) (95%CI: 1.51–2.27)	1.8 (40) (95%CI: 1.6–1.95)	5.48 (3.00–6.20)	4.6 (2.6–6.9)	27.9 (18.4) (95%CI: 23.1–33.8)	30.3 (45) (95%CI: 27.2–33.9
Food Effect [11]	200	Moderate-fat	1.43 (35.6) (95%CI: 1.2–1.7)	1.0 (42) (95%CI: 0.96–1.1)	5.00 (2.00–8.00)	5.25 (3.0–8.4)	41.0 (42.2) (95%CI: 34–49.6) ^b^	31.9 (48) (95%CI: 29.2–35)
		High-fat	1.08 (38.6) (95%CI: 0.91–1.28)		5.00 (1.50–12.0)		36.9 (35.9) (95%CI: 31.5–43.3) ^b^	
		Fasted	0.35 (84.4) (95%CI: 0.23–0.55)	0.36 (40) (95%CI: 0.33–0.39)	4.0 (2.03–24.0)	4.53 (1.9–6.3)	13.5 (58.2) (95%CI: 9.1–19.8) ^b^	10.9 (47) (95%CI: 10.0–12)
AAFE			1.23		1.15		1.30	

Note: * Geometric Mean (geometric%CV); ** Median (range); *** Day14; **** Day7 (part2); ^a^ clinical studies conducted only with moderate-fat meal; ^b^ AUC0-∞; ^c^ AUC0-τ, AUC from pre-dose to the end of the dosing interval at steady state.

## Data Availability

The data presented in this study are available in this article (and Appendix A).

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
