# Peer review of "Integrating In Vitro Biopharmaceutics into Physiologically Based Biopharmaceutic Model (PBBM) to Predict Food Effect of BCS IV Zwitterionic Drug (GSK3640254)"

_pharmaceutics, 2023, doi:10.3390/pharmaceutics15020521_

Round 1

Reviewer 1 Report

One of the most impressive papers I've ever seen. I just encountered few problems of technical nature concerning figures and tables and the text flow around them but that's technicality. I think it'd be beneficial to add the separate glossary of abbreviations. Once again my congratulations on this paper.

Author Response

We would like to thank the reviewer for the kind words. We have fixed the technical issues and we hope that this time is more readable.

Reviewer 2 Report

MANUSCRIPT: 2185216

TITLE: Integrating in vitro biopharmaceutics into Physiologically Based Biopharmaceutic Model (PBBM) to predict Food effect of 3 BCS IV zwitterionic drug (GSK3640254)

The manuscript 2185216 “Integrating in vitro biopharmaceutics into Physiologically Based Biopharmaceutic Model (PBBM) to predict Food effect of 3 BCS IV zwitterionic drug (GSK3640254)”, presents an interesting study that shows us how PBBM models predict food effect of BCS class IV drugs.

The work is well structured, well planned and the research is competently carried out, the methodology was adequate to the research. The literature cited is adequate and most of the papers cited are from the last five years.

The results and discussion are properly discussed. Conclusions are presented according to the results obtained.

However, some questions remain to be clarified and solved and the manuscript in the current form must be revised in minor several points as follows comments:

1. Line 17 - Please change “de-scribed” to “described”.

2. Line 202 – TFA is not described in section 2.1 List of chemicals. Please place in section 2.1 this chemical TFA.

3. Line 212 - Please - Latin expressions should be written in italics, I would recommend writing the expression "et al" throughout the manuscript in italics.

4. Table 2 – In the Total BS (mM) column, the abbreviation BS appears for the first time. Please write in full what BS means or previously describe in the text in full BS.

5. Secção 2.7 - Prediction accuracy and statistical analysis. Please indicate the (n) – number of replicates in each parameter to verify statistically significant differences by ANOVA.

6. Figure 3 and figure 4 captions – please indicate in addition to the mean ± SD the n= number of replicates in each parameter.

7. Table 4 – Please what does A, B, M.B. and s.d. means. Please describe these acronyms in the manuscript text or in footnote in Table 4.

8. Table 5 – The table is repeated in the manuscript and is not previously referred to in the text, please correct the situation.

9. Table 5 – Please change “hr” by “h” and put in footnote what CMax, AUC, Vss, CL, T1/2 means.

10. Results (Pages 15 a 24) - Please, it is recommended to reorganize the manuscript when presenting the results between pages 15 and 24, namely:

a) Figures and tables must be previously referred in the text of manuscript.

b) The captions of figures and tables must be placed correctly in each one of them.

c) Tables that are in duplicate and in triplicate in the manuscript must be removed.

d) The results in the tables must be presented as mean ± standard deviation and indicate the n (number of replicates).

11. References must be presented in accordance with the Reference List and Citations Style Guide for MDPI Journals. Please download the full MDPI Reference List and Citations Style Guide at MDPI | Reference List and Citations Style Guide and proceed as per the document and present the references as per the document and include the doi and/or ISBN.

Author Response

We would like to thank the reviewer for the comments and suggestions...below we provide point-by-point response.

Reviewer 2:

The manuscript 2185216 “Integrating in vitro biopharmaceutics into Physiologically Based Biopharmaceutic Model (PBBM) to predict Food effect of 3 BCS IV zwitterionic drug (GSK3640254)”, presents an interesting study that shows us how PBBM models predict food effect of BCS class IV drugs.

The work is well structured, well planned and the research is competently carried out, the methodology was adequate to the research. The literature cited is adequate and most of the papers cited are from the last five years.

The results and discussion are properly discussed. Conclusions are presented according to the results obtained.

However, some questions remain to be clarified and solved and the manuscript in the current form must be revised in minor several points as follows comments:

  1. Line 17 - Please change “de-scribed” to “described”.

Authors: Revised

  1. Line 202 – TFA is not described in section 2.1 List of chemicals. Please place in section 2.1 this chemical TFA.

Authors: TFA was added in section 2.1

  1. Line 212 - Please - Latin expressions should be written in italics, I would recommend writing the expression "et al" throughout the manuscript in italics.

Authors: Revised

  1. Table 2 – In the Total BS (mM) column, the abbreviation BS appears for the first time. Please write in full what BS means or previously describe in the text in full BS.

Authors: Revised…we added an explanation of this abbreviation in the footnote of Table 2.

  1. Secção 2.7 - Prediction accuracy and statistical analysis. Please indicate the (n) – number of replicates in each parameter to verify statistically significant differences by ANOVA.

Authors: The (n) number has been added to that section. See revised manuscript

  1. Figure 3 and figure 4 captions – please indicate in addition to the mean ± SD the n= number of replicates in each parameter.

Authors: N number was added to the captions of figure 3 and 4

  1. Table 4 – Please what does A, B, M.B. and s.d. means. Please describe these acronyms in the manuscript text or in footnote in Table 4.

Authors: An explanation of the acronyms have been added into a footnote in Table 4.

  1. Table 5 – The table is repeated in the manuscript and is not previously referred to in the text, please correct the situation.

Authors: The Table 5 is already referred into to the text. Please see line 526.

  1. Table 5 – Please change “hr” by “h” and put in footnote what CMax, AUC, Vss, CL, T1/2 means.

Authors: “hr” by “h” changed and a foot note has been added for CMax, AUC, Vss, CL and T1/2…

  1. Results (Pages 15 a 24) - Please, it is recommended to reorganize the manuscript when presenting the results between pages 15 and 24, namely:

  1. a) Figures and tables must be previously referred in the text of manuscript.

Authors: All the figures and tables have been previously referred in the text. Probably due to the technical issues with the figures, might not be clear the position of the figures with respect to their reference in the text

  1. b) The captions of figures and tables must be placed correctly in each one of them.

Authors: Yes, we solved this technical issue.

  1. c) Tables that are in duplicate and in triplicate in the manuscript must be removed.

Authors: All the tables are unique and have been reported only once in the manuscript.

  1. d) The results in the tables must be presented as mean ± standard deviation and indicate the n (number of replicates).

Authors: The results in Table 8 are presented based on how the clinical data have been published. So, we used the same format. In Table 6 & 7 we reported only the mean values as SIVA provides only a single average predicted value. We updated Table 5 to include %CV as reported in the clinical study. The n number is already reported within the title of the Table 5.

  1. References must be presented in accordance with the Reference List and Citations Style Guide for MDPI Journals. Please download the full MDPI Reference List and Citations Style Guide at MDPI | Reference List and Citations Style Guide and proceed as per the document and present the references as per the document and include the doi and/or ISBN.

Authors:  We have used the style provided by the journal located in this link MDPI | Reference List and Citations Style Guide. The papers have been referenced automatically to the manuscript using this style…we think the ISBN and or doi will be reported by the journal later. We can’t add doi on the manuscript, although, are included into the endnote library we have. So, what you see as reference format is based on the citation style we downloaded from the journal’s website…

Reviewer 3 Report

Please look at the comments (pdf doc).

Author Response

We would like to thank the reviewer and the comments and suggestions. Below we provide a point-by-point response.

Reviewer 3:

Review Report:

Thank you for providing me the opportunity to review this informative manuscript. While

reading the manuscript, I have come across the following observations:

Summary:

The findings of an experiment were demonstrated in this manuscript that was based on the

development of PBBM model using clinical data to predict the food effect of a BCS class IV

zwitterion drug (GSK3640254).

The study said that, the positive food effect was attributed to micelle mediated enhanced

solubility and permeability. The media containing oleic acid and cholesterol in fasting and fed state made the model to appropriately capture the food effect intensity. Additionally, I found the technology interesting but not novel that can help resolving issues with nutrient-drug interaction as because, nowadays, due to the component present in food, the drug that is administered to resolve disease consequences could not reach at its peak success due to unwanted interaction between food and drug…

Author: We agree with the reviewer that more need to be done to better understand and model food-drug interactions. However, there is bigger problem which has to do with how we develop PBBM models, how we integrate the in vitro data and how we make the right assumptions/adjustments to the models. This is what we tried to do and to address the current limitations of the PBPK platforms, to address the improvements need to bedone.

Issues: The overall writing and presentation of the manuscript need to improve to a big deal.

Figure 1 and 2 are not clear (overlap/blur) and the authors could use a better software for more visibility. No legands/footnotes were used in tables in the result section.

In the subheading,

3.3.1.2. Dissolution and permeability, in line 549, a nice figure was used, however, that could be placed at the end of paragraph with appropriate footnote and should also be mentioned in the write up.

Even for figure 6 in line 570, the title should be clearly written on the bottom of the

figure. I did not find any figure 7 (mentioned nicely in line 612). Overall there are too many

tablels and figures, specially in the result section and they are not put with care.

Authors: We would like to apologise for the technical issues. We faced the same issue when we were writing the manuscript and tried to upload it to the journal. We worked more on the settings, and we hope that the issue is not there this time and the quality of the manuscript has been improved.

Reviewer 4 Report

Comments to Author:

Minor typo errors (should be checked through all manuscript)

Page 2, line 46: the meaning of the abbreviation GI should be written down when it is first mentioned in the text (…. gastrointestinal tract (GI tract) ….).

Page 9, line 281: remove the brackets when using the abbreviation (PSA) in the text.

Page 10, line 363: The AAFE is abbreviation and should be written in brackets, while the meaning of the abbreviation must be written for the first time before the abbreviation as follow: The average absolute fold error (AAFE) was calculated …. .

Page 15, line 552: Only abbreviation PSA should be used, without listing the meaning of the abbreviation, because it was already used previously.

Section 2 Materials and Methods

Page 4, Figure 1: The legend font in Figure 1B is too small.

Page 5, line 206: please specify which external standard you used for HPLC analysis.

Section 3 Results

Subsection 3.1.1. Fasted

Explain why you chose only one pH to conduct fasted media study while you used three pH values for the fed media application study?

Subsection 3.3.1.2. Dissolution and permeability

Does the picture in this subsection belong in the supplementary materials? Where is the picture description? Additionally, the font is too small. Or is this the Figure 6?

Also Figure 7 is missing or you have a lag in the images and their descriptions. Please, correct confusion with figures throughout the whole manuscript.

Section Conclusion

The conclusion should be more impressive. The contribution of the obtained results to the science or possible applications should be more emphasized. Additionally, the authors should highlight their novelties gained from the study compared to similar studies.

Author Response

We would like to thank the reviewer for the comments and suggestions. Below we provide a point-by-point response.

Reviewer 4

Minor typo errors (should be checked through all manuscript)

Page 2, line 46: the meaning of the abbreviation GI should be written down when it is first mentioned in the text (…. gastrointestinal tract (GI tract) ….).

Authors: Revised

Page 9, line 281: remove the brackets when using the abbreviation (PSA) in the text.

Authors: Revised

Page 10, line 363: The AAFE is abbreviation and should be written in brackets, while the meaning of the abbreviation must be written for the first time before the abbreviation as follow: The average absolute fold error (AAFE) was calculated …. .

Authors: Revised

Page 15, line 552: Only abbreviation PSA should be used, without listing the meaning of the abbreviation, because it was already used previously.

Authors: Revised

Section 2 Materials and Methods

Page 4, Figure 1: The legend font in Figure 1B is too small.

Authors: We increased the size of the figure and we printed out to check if it’s readable and it is. We hope that’s sufficient.

Page 5, line 206: please specify which external standard you used for HPLC analysis.

Authors: we rephrased this to translate an internal definition to a general accepted terminology. Practically, we generated a calibration curve of GSK254 and was used to quantify the area of the peaks in HPLC in each sample.

Section 3 Results

Subsection 3.1.1. Fasted

Explain why you chose only one pH to conduct fasted media study while you used three pH values for the fed media application study?

Authors: Initially we used three media, only taurocholate (TC), then TC and phospholipids (PC) and then TC+PC+cholesterol) and we changed the pH. The results showed no meaningful impact of pH on the solubility of the drug at different composition of fasted media. It was mainly the low solubilization capacity of the media, i.e., low bile salt, phospholipids and cholesterol concentration. This is the reason, as we mentioned in the manuscript, the in fasted state is solubility limited process…so exploring further and performing more solubility experiments using different media composition at different pH values had no meaning. So, we used the average pH (6.5) and we explored in more details the impact of the composition of the media on the solubility of the drug, using mixture bile salts with or without PC and cholesterol.

In fed media the pH had a bigger impact, as changes in the ionization of the drug led to changes in the affinity of the drug to micelles and hence changes in the solubility of the drug. In this case, although the solubilization capacity of the media was much higher compared to fasted…it was the ionization of the drug that dictated the degree of the partition of the drug to the mixed micelles.

We added an extra sentence in the manuscript to address this. Thank you for mentioning it.

Subsection 3.3.1.2. Dissolution and permeability

Does the picture in this subsection belong in the supplementary materials? Where is the picture description? Additionally, the font is too small. Or is this the Figure 6?

Authors: The picture is actually figure 6. We improve words’ settings to avoid these technical issues.

Also Figure 7 is missing or you have a lag in the images and their descriptions. Please, correct confusion with figures throughout the whole manuscript.

Authors: Yes, we solved these technical issues. Thank you for your suggestion!

Section Conclusion

The conclusion should be more impressive. The contribution of the obtained results to the science or possible applications should be more emphasized. Additionally, the authors should highlight their novelties gained from the study compared to similar studies.

Authors: Thank you for the suggestion. We have extended the conclusion section, including some key learnings from this work. Please see these additions in the revised manuscript.